# STOCHASTOK: IMPROVING FINE-GRAINED SUBWORD UNDERSTANDING IN LLMS

**Anya Sims**
FLAIR, University of Oxford
anya.sims@stats.ox.ac.uk

**Thom Foster**
FLAIR, University of Oxford

**Tuan-Duy H. Nguyen**
National University of Singapore

**Klara Kaleb**
University of Oxford

**Joseph Lee**
University of Oxford

**Jakob N. Foerster**
FLAIR, University of Oxford

**Yee Whye Teh**
University of Oxford

**Cong Lu**
University of British Columbia

## ABSTRACT

Subword-level understanding is integral to numerous tasks, including understanding multi-digit numbers, spelling mistakes, abbreviations, rhyming, and wordplay. Despite this, current large language models (LLMs) still struggle disproportionally with simple subword-level tasks like *How many 'r's in 'strawberry'?*. A key factor behind these failures is tokenization which obscures the fine-grained structure of words. Current alternatives, such as character-level and dropout tokenization methods, significantly increase computational costs and provide inconsistent improvements. In this paper we revisit tokenization and introduce STOCHASTOK, a simple, efficient stochastic tokenization scheme that randomly splits tokens during training, allowing LLMs to 'see' their internal structure. Our experiments show that pretraining with STOCHASTOK substantially improves LLMs' downstream performance across multiple subword-level language games, including character counting, substring identification, and math tasks. Furthermore, STOCHASTOK's simplicity allows seamless integration at any stage of the training pipeline; and we demonstrate that post-training with STOCHASTOK can instill improved subword understanding into existing pretrained models, thus avoiding costly pretraining from scratch. These dramatic improvements achieved with a minimal change suggest STOCHASTOK holds exciting potential when applied to larger, more capable models. Code open-sourced at: github.com/anyasims/stochastok.

## 1 INTRODUCTION

Large language models (LLMs) have achieved remarkable progress on a wide range of tasks (Achiam et al., 2023; Team et al., 2023; Dubey et al., 2024). However, their reliance on tokenization (Sennrich et al., 2016) obscures how humans naturally perceive language. For example, while humans see 'book' and 'cook' as differing by a single letter, when training LLMs, we always treat these words as distinct token IDs[1]. This makes subword-level tasks such as *How many 'r's in 'strawberry'?* difficult, even for current state-of-the-art LLMs. Whilst some advanced reasoning models, such as OpenAI's o1 (Jaech et al., 2024), have recently started to show promise, it has required a vast increase in model size and training complexity that seems disproportionate to the simplicity of such questions. In the arts, this shortcoming impacts wordplay, rhyming, and understanding etymology, while in the sciences, it is needed for handling multi-digit numbers, chemical formulae, and mathematical equations. Moreover, these failures highlight a fundamental inability of LLMs to understand how humans perceive language, an essential aspect of effective communication with humans.

This limitation in standard tokenizers has motivated research into stochastic tokenization, where 'stochastic tokenization' refers to methods in which the same text may be encoded as multiple possible

---

[1]e.g., 'book'=3092 and 'cook'=171691 in the GPT-4o and GPT-4o mini models (Hurst et al., 2024).

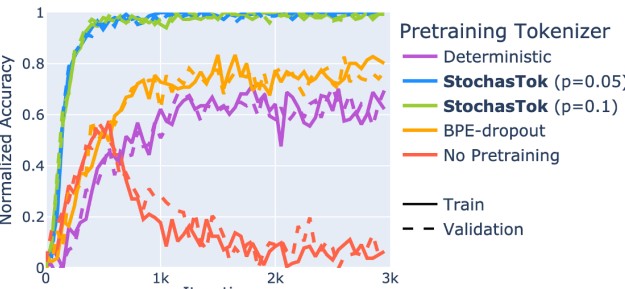

Figure 1: STOCHASTOK pretraining allows the learned representations to capture the fine-grained details of how humans 'see' language. This is demonstrated as models pretrained with STOCHASTOK can be finetuned to answer language game questions with no compromise to ability in other domains.

token sequences. A well-known existing method is BPE-dropout (Provilkov et al., 2020), which adds randomness by skipping BPE merge steps. In this work, we propose a simpler, more flexible, and more effective alternative: rather than modifying the original tokenization process, we instead allow LLMs to directly 'see' inside tokens by randomly splitting them into equivalent pairs of smaller tokens with some small probability.

Our experiments show that adding this minimal additional preprocessing step significantly alters the model's representations, allowing them to capture subtoken-level morphological structure. Compared to prior stochastic tokenization methods (Provilkov et al., 2020; Kudo, 2018), we find STOCHASTOK to be significantly more effective, while also having strong practical advantages of being faster, simpler, compatible with any base tokenizer, and applicable post-hoc to existing pretrained models.

We demonstrate three main results. Firstly, language models pretrained with STOCHASTOK quickly adapt to near-perfect accuracy on several language game tasks (such as 'Which word has the most e's?' or 'Which word is the shortest?'), while models pretrained with deterministic tokenization or BPE-dropout struggle (see Figure 1). We test this on two sets of language game tasks: (1) LangGame - our novel set of subword understanding tasks, and (2) the CUTE benchmark of language manipulation tasks (Edman et al., 2024). Secondly, we show that STOCHASTOK enables models to grok multi-digit addition, a dramatic change in learning behavior compared to BPE-dropout or deterministically trained models (Lee et al., 2023). Thirdly, since STOCHASTOK is compatible with existing pretrained models, we demonstrate that it can be used to 'retrofit' larger existing pretrained models with improved subword understanding, thus mitigating the need to pretrain from scratch. In summary, STOCHASTOK provides a stark performance improvement with minimal cost or implementation changes, and we believe our results at the modest scale have potential for major impact on LLM ability when used to pretrain or finetune larger, more capable models.

## 2 BACKGROUND

Tokenization (Sennrich et al., 2016)—the process of converting raw text into tokens—serves two essential roles in the LLM pipeline. Firstly, it converts text into a sequence of integers to enable processing by the LLM. Secondly, it compresses sequences of characters into shorter sequences of tokens, which increases both performance and computational efficiency.

**Standard Deterministic Tokenization.** A tokenizer consists of two main components: a vocabulary, and an encoding function for converting text into a sequence of token IDs. The decoding procedure shared by all tokenizers simply maps token IDs back to text strings. For instance, with vocabulary `{0:The,1:_c,2:at,3:_s,...}`, the sequence `[0,1,2,3,2]` decodes to `'The_cat_sat'`.

The main tokenizers are Byte-Pair Encoding (BPE; Sennrich et al. (2016)) and Unigram (Kudo, 2018). **BPE** is constructed by starting with individual character tokens and iteratively merging the most frequent adjacent token pairs in a training dataset, yielding a fixed-size vocabulary and a hierarchical set of merge rules. For encoding, text is initially split into character-level tokens, and the merge rules are applied repeatedly until no further merges are possible. In contrast, **Unigram** starts with a large candidate vocabulary and iteratively prunes tokens that least increase the dataset's log-likelihood under a unigram model, using the Viterbi (Viterbi, 1967) and EM (Dempster et al., 1977) algorithms to compute and optimize token probabilities. For encoding, the tokenization with the highest probability under the learned unigram model is selected using the Viterbi algorithm. BPE

is currently the choice of most SOTA LLMs (Groeneveld et al., 2024; Dubey et al., 2024; Team et al., 2024; Jiang et al., 2023; Abdin et al., 2024; Guo et al., 2025; Yang et al., 2024; Biderman et al., 2023) due to having much lower memory requirements than Unigram.

**Stochastic Tokenization.**    BPE and Unigram are deterministic tokenizers, meaning the same input text always produces the same tokenization.    We define stochastic tokenization as any tokenizer whose encoding function may produce multiple alternative tokenizations for the same input. With `vocab={0:e, 1:x, 2:a, 3:m, 4:p, 5:l, 6:exam, 7:ple, 8:example}`, for example, the word 'example' might be mapped to any of `[8]`, `[6,7]`, `[0,1,2,3,4,5,0]`, `etc.`, since the decoding procedure (identical to deterministic tokenizers) will map each of these back to the text 'example'.

The two main prior stochastic tokenization methods are Subword Regularization and BPE-dropout. **Subword Regularization** (Kudo, 2018) extends Unigram by sampling from alternative tokenizations according to learned unigram model probabilities. However, this adds complexity and computational overhead to the already expensive Unigram procedure, and introduces intricacies involving overlapping candidates, beam tuning, and numerical stability. **BPE-dropout** (Provilkov et al., 2020) introduces stochasticity by randomly omitting some merge operations of BPE during encoding. Unfortunately, this results in a different vocabulary from the original BPE tokenizer,[2] preventing easy application to pretrained models. It also incurs additional drawbacks such as higher computational costs, unwanted tokenization dependence on text length, and is only compatible with BPE. In our experiments we therefore compare to BPE, the defacto standard in SOTA LLMs, and BPE-dropout, the only prior BPE-compatible stochastic variant (see Section 8).

## 3 STOCHASTOK

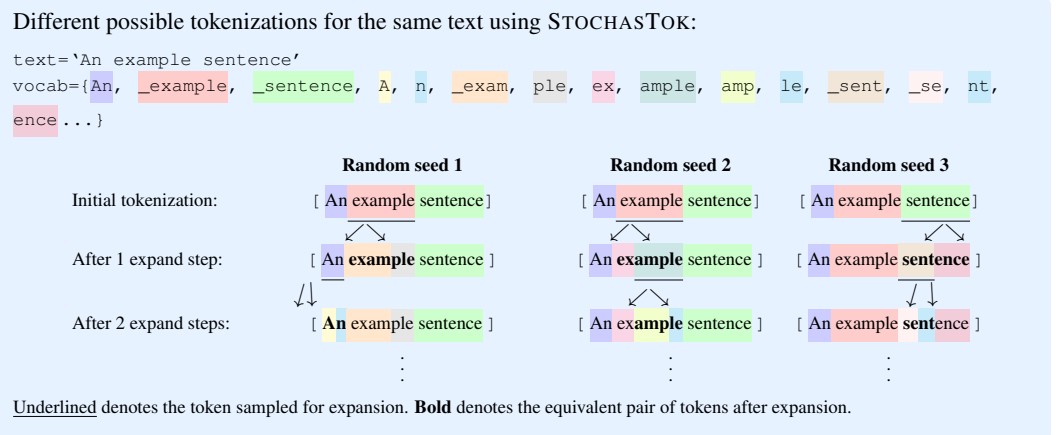

Different possible tokenizations for the same text using STOCHASTOK:

```
text='An example sentence'
vocab={An, _example, _sentence, A, n, _exam, ple, ex, ample, amp, le, _sent, _se, nt,
ence ...}
```

Underlined denotes the token sampled for expansion. **Bold** denotes the equivalent pair of tokens after expansion.

Figure 2: STOCHASTOK involves iteratively sampling tokens to 'expand' into equivalent pairs of tokens in the vocabulary, resulting in multiple possible tokenizations for the same text. The exposure to alternative tokenizations enables LLMs to naturally learn about the fine-grained subtoken-level morphological composition of tokens.

In this section, we describe STOCHASTOK, a simple, lightweight, stochastic tokenization scheme that, unlike prior work, is compatible with any base tokenizer or pretrained model.

STOCHASTOK involves two steps:

1. Tokenize with the base tokenizer to get a list of `token_ids`.

2. Iteratively apply 'expand' steps in which a token is sampled at random and (if possible) split into a pair of equivalent tokens in the vocabulary (as depicted in Figure 2). This is repeated for $p \cdot$ `len(token_ids)` iterations, where $p$ is a hyperparameter.

---

[2]In BPE, intermediate tokens not present in the final tokenized training dataset are removed from the vocabulary, meaning BPE-dropout can produce tokens outside the original vocabulary.

In Step 2, if no equivalent pairs of tokens exist for the sampled token (e.g., if the token is already a single character), then the expand step is skipped. Full pseudocode is given in Section A.3, and further illustrative examples in Section A.4. Through this repeated token re-segmentation the model is exposed to many alternative tokenizations; for example, the word [example] may appear in the dataset as any of: [example], [exam|ple], [ex|ample], [ex|am|ple], [e|x|am|ple], etc, thus allowing it to learn the fine-grained structure of words.

STOCHASTOK has several *practical* advantages:

- **Cheap and efficient.** STOCHASTOK is considerably cheaper than existing methods both in terms of memory and compute. Rather than re-tokenizing from scratch, data can be tokenized once and cheaply expanded for varying numbers of 'expand steps' to achieve different levels of stochasticity.
- **Compatible with any tokenizer.** Unlike BPE-dropout or Subword Regularization, STOCHASTOK can be applied to any base tokenizer (BPE, Unigram, WordPiece, etc.) without requiring any knowledge of the base tokenizer itself.
- **Extremely simple.** STOCHASTOK is simply a lightweight post-processing step after tokenization. Everything else—including the training loop—remains unchanged.
- **Preserves original vocabulary.** Perhaps most significantly, STOCHASTOK maintains the original tokenizer vocabulary, thus allowing straightforward application to any stage of the LLM pipeline. In Section 4 we apply STOCHASTOK during pretraining and switch it off seamlessly for downstream finetuning, while in Section 6, we apply STOCHASTOK after pretraining to instill subword understanding into existing pretrained models.
- **Robust to hyperparameter choice.** STOCHASTOK is robust to hyperparameter choice (see Figure 5) and hence does not require careful tuning. By default we use $p = 0.1$, and show similar effectiveness with $p = 0.05$ and other values.

In the following sections, we demonstrate STOCHASTOK's *empirical* advantages. Firstly, we show that pretraining with STOCHASTOK dramatically improves downstream performance on language game tasks, while being *(a)* extremely robust to hyperparameter choice and *(b)* exhibiting out-of-distribution generalization properties (Section 4). Next, we examine math tasks and find that models trained with STOCHASTOK quickly grok multi-digit addition—and moreover generalize to unseen test tokenization schemes—whereas models trained with existing tokenizers struggle, even when tested with the matching tokenizer (see Section 5). We then apply STOCHASTOK to existing pretrained models and demonstrate that it can be used to 'retrofit' improved subtoken understanding into larger deterministically pretrained models (Section 6). Finally, we provide insights into the internal mechanisms of STOCHASTOK-trained models compared to models trained with standard tokenization (Section 7).

## 4 STOCHASTOK PRETRAINING ENABLES SUCCESS IN LANGUAGE GAMES

**Setup.** In this section, we look at the effect of STOCHASTOK when applied during pretraining. We build on the baseline open-source setup of Hillier et al. (2024) (a 50M-parameter model, using GPT-2 BPE tokenizer, trained on the OpenWebText dataset—see Section C.1 for full details). We compare four models: (1) Pretrained with standard deterministic tokenization, (2) Pretrained with STOCHASTOK, (3) Pretrained with BPE-dropout, and (4) No pretraining. Firstly, in Figure 3, we verify that STOCHASTOK requires no compromise in original language modeling performance (see Section C.1 for benchmark details).

| | Task | Question | Answer |
|---|---|---|---|
| 1 | Letter | Which word has the most letter 'n's? The options are: [ reason, step, continent, their]. | continent |
| 2 | Contains | Which choice contains 'ec'? The option words are: [ was, children, require, check]. | check |
| 3 | Starts | Which option string starts with 'mo'? The available options: [ case, ask, month, event]. | month |
| 4 | Ends | What option word ends with 'ad'? The option words are: [ cost, lead, south, sun]. | lead |
| 5 | Longest | Which string is the longest? The available choices: [ wild, dear, had, section]. | section |
| 6 | Shortest | Which is the shortest? The possible option words: [ thought, job, circle, nothing]. | job |

Table 1: We introduce 'LangGame,' a novel dataset consisting of six question types testing fine-grained subword-level understanding.

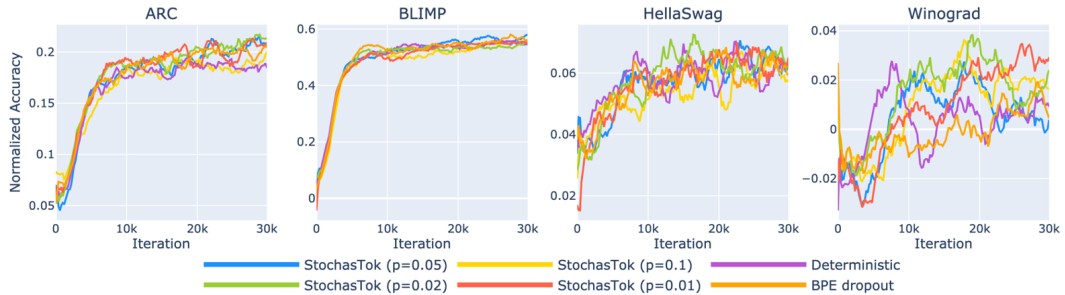

Figure 3: We first verify that STOCHASTOK does not compromise test performance across a wide variety of standard language understanding benchmarks.

**Performance on Language Game Tasks.** We now finetune each of the base models above on two sets of language game tasks: (1) LangGame, and (2) CUTE. LangGame is a novel dataset consisting of six different tasks, including identifying word lengths, substrings, and individual letters. Examples are shown in Table 1, and additional detail is given in Section B.1. The CUTE benchmark contains further language manipulation tasks (Edman et al., 2024) (see Section B.2 for examples). **Critically, each model is finetuned identically, using deterministic BPE tokenization.**

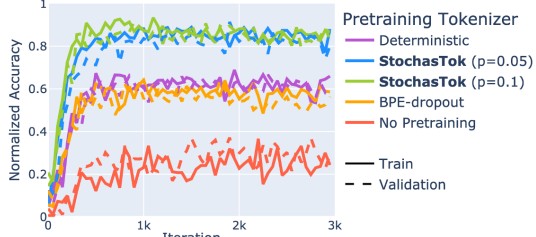

Figure 4: Pretraining with STOCHASTOK enables significantly higher performance on the CUTE language manipulation tasks (in addition to the LangGame tasks—see Figure 1). (For 'normalized accuracy,' 0 is random guessing and 1 is perfect.)

Figure 1 shows performance on the LangGame questions. We observe that the models pretrained with STOCHASTOK quickly achieve near-perfect accuracy, while the models pretrained with deterministic tokenization or no pretraining are unable to reach high accuracy. This suggests that, as well as the token-level structure learned with deterministic tokenization, STOCHASTOK enables models to additionally capture subtoken-level fine-grained morphological structure. The prior method of BPE-dropout gives some of the benefits of stochastic tokenization, but still performs significantly worse than STOCHASTOK, in addition to being significantly more complex. In Figure 4, we see that STOCHASTOK gives a similar stark performance difference on the CUTE language manipulation benchmark, thus giving further evidence that STOCHASTOK significantly changes the representations of the model to enable fine-grained character-level manipulation.

**Robust to Hyperparameter Choice and OOD Questions.** In addition to significant performance increases on both language game benchmarks, we find that the benefits of stochastic tokenization are robust over an order of magnitude range of the hyperparameter (see Figure 5). Furthermore, we find that this skill is learned in a way that enables the model to generalize to a set of holdout language game question types in which the train/validation questions all involve identifying substrings/prefixes/suffixes where the substring/prefix/suffix is always less than or equal to half the answer length, while in the holdout set the substring/prefix/suffix is always longer than half the

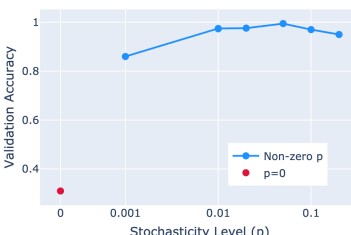

Figure 5: STOCHASTOK is effective over a wide range of stochasticity levels (log x-scale), meaning it is robust to hyperparameter choice.

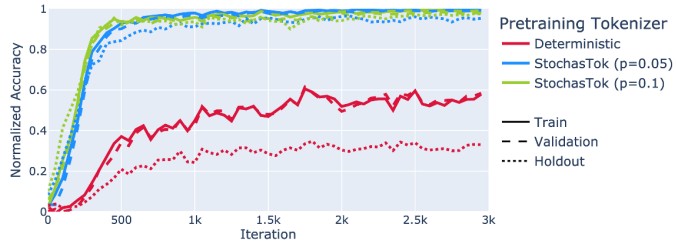

Figure 6: Models pretrained with STOCHASTOK successfully generalize to out-of-distribution language game questions, while those pretrained deterministically exhibit a significant generalization gap (and a much lower in-distribution performance).

answer length. In Figure 6, we observe that models pretrained with stochastic tokenization generalize near-perfectly while the deterministic tokenization-pretrained equivalent has a significant generalization gap in addition to a much lower in-distribution performance.

**Transfers to Larger Models.** Next, we verify that these findings transfer to larger settings by applying STOCHASTOK to the `modded-nanogpt` baseline (Jordan et al., 2024a). This setup has a different architecture and model size of GPT-2 with 275M parameters, a different training dataset (FineWeb Penedo et al. (2024)), and a different optimizer (Muon Jordan et al. (2024b)). In Figure 7, we see that STOCHASTOK gives a similar performance benefit in this larger setting, suggesting that STOCHASTOK scales to larger models.

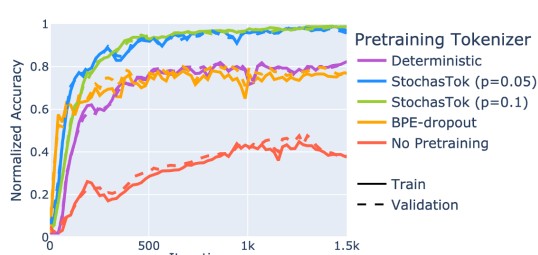

Figure 7: STOCHASTOK also enables improved LangGame performance in larger models.

## 5 STOCHASTOK ENABLES LLMS TO GROK MATH TASKS

In addition to language game-type tasks, tokenization also poses difficulties in learning math, due to obscuring the relation between numbers, for example in GPT-4o (Hurst et al., 2024), the numbers '2', '20', '200', '201' are tokenized as `17`, `455`, `1179`, `667` respectively. This poses such a significant additional difficulty for language models that prior works commonly use tricks like adding '.'s between every character (to force tokenization to keep each digit separate), or using custom character-level tokenizers for digits to sidestep the issue (Zhang et al., 2024; Power et al., 2022; Lee et al., 2023).

We hypothesize that since STOCHASTOK improves sub-token level awareness, it may also help in learning multi-digit math tasks. To test this, we train on the task of multi-digit addition starting from the 50M-parameter setup in Hillier et al. (2024). Examples of the questions are given in Section B.3. We compare the performance of models trained with: (1) standard deterministic tokenization, (2) BPE-dropout, (3) STOCHASTOK, and (4) character-level tokenization. In Figure 8, for each of the four models we plot the accuracy with the question tokenized with each of the four methods.

In Figure 8 *left*, we see—as expected—that the character-level-trained model quickly achieves near-perfect accuracy when the questions are tokenized character-wise (and gets near-zero accuracy when the questions are tokenized differently). In Figure 8 *middle-left* and *middle-right*, we see that the models trained with standard deterministic tokenization and BPE-dropout struggle to grok the task, appearing to slowly learn examples with the accuracy increasing linearly, even with the matching question tokenization. By contrast, **in Figure 8 *right*, the model trained with STOCHASTOK quickly groks the task and reaches near-perfect accuracy, not just when the question is tokenized with the matching tokenizer, but also when the question is tokenized with any of the other three tokenizers that were unseen during training**. This suggests that STOCHASTOK significantly enhances a model's ability to understand relationships between multi-digit numbers.

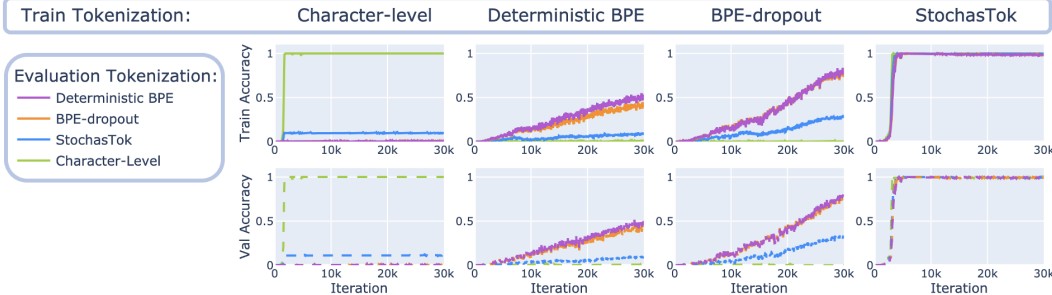

Figure 8: **STOCHASTOK allows models to grok multi-digit addition**. Unlike training with character-level or deterministic BPE tokenizers, training with STOCHASTOK achieves near-perfect validation accuracy even when tested with questions tokenized with methods not seen during training.

## 6 STOCHASTOK CAN INSTILL SUBWORD UNDERSTANDING INTO EXISTING PRETRAINED MODELS

Pretraining is often prohibitively expensive. In this section, we therefore investigate whether STOCHASTOK can be used to instill improved subword understanding into models that have already been pretrained with an alternative tokenization method, offering a more cost-effective alternative to full retraining from scratch. For our first experiment, we start with the 50M-parameter model from Section 4, which was trained for 30k iterations on OpenWebText using deterministic BPE. We call this the 'base model.' We then continue to train for an additional 2k iterations on OpenWebText with STOCHASTOK tokenization, which we refer to as continued pretraining (CPT). As a control, we also perform CPT with standard deterministic BPE. As before, we then try finetuning on the LangGame tasks. In Figure 9, we show that a small amount of CPT is sufficient to enable the models to fit the language game questions near-perfectly, significantly higher than all of the controls. This suggests that the 2k steps of CPT with STOCHASTOK were effective in instilling subword understanding into the pretrained model.

**Larger Pretrained Models.** Next, we test this on a larger open-source model. In Figure 10, we compare the ability of GPT-2 (Radford et al., 2019) to fit the language game tasks with (1) no additional pretraining, (2) 7k iterations of CPT with deterministic BPE, and (3) 7k iterations of continued pretraining with STOCHASTOK. CPT with deterministic BPE has no effect on the ability to learn the LangGame tasks, whilst STOCHASTOK again allows the model to reach significantly higher accuracy.

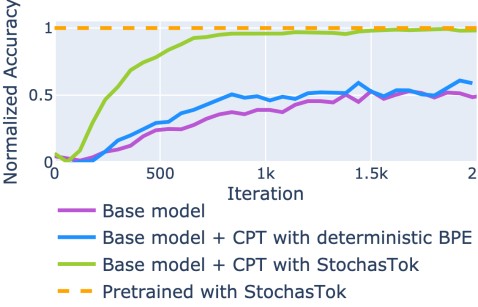

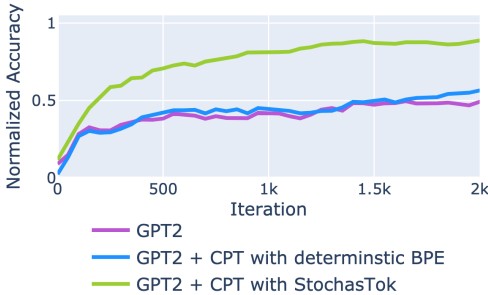

Figure 9: A small amount of continued pretraining (CPT) with STOCHASTOK significantly improves subword awareness in the 50M-parameter deterministically-pretrained baseline.

Figure 10: The effectiveness of STOCHASTOK in continued pretraining (CPT) transfers to the larger setting, enabling the pretrained GPT-2 model to fit language game tasks.

## 7 ANALYSIS

Finally, we present an analysis of how STOCHASTOK enables the improvements in subword-level understanding. In Figure 11, we show completions when prompted with different tokenizations of the same prompt. We find that—as expected—the responses from the model trained with STOCHASTOK are much more consistent across different prompt tokenizations, while the standard tokenization-trained model quickly breaks down when exposed to alternative tokenizations.

| Different prompt tokenizations | Deterministic Training | StochasTok Training |
|---|---|---|
| The chef sighed and put the | knife in his mouth . " | dish on the table . |
| The chef sigh ed and p ut the | words . ◆ ◆ | dish on the table . |
| T he ch ef sighed a nd put t he | on the table and said , | m oust ache on the |
| The chef sighed and pu t th e | au , and then he turned | dish on the table . |
| Th e chef sighed and p ut the | words . " I 'm not | dish on the table . |
| The chef s ighed an d put the | dish on the table and then | dish on the table . |

Figure 11: Generations given multiple different tokenizations of the same prompt. We find the STOCHASTOK-trained model to be more consistent, while the standard-trained model breaks down when prompted with alternative tokenizations, showing STOCHASTOK improves tokenization robustness. More examples are provided in Section D.1.

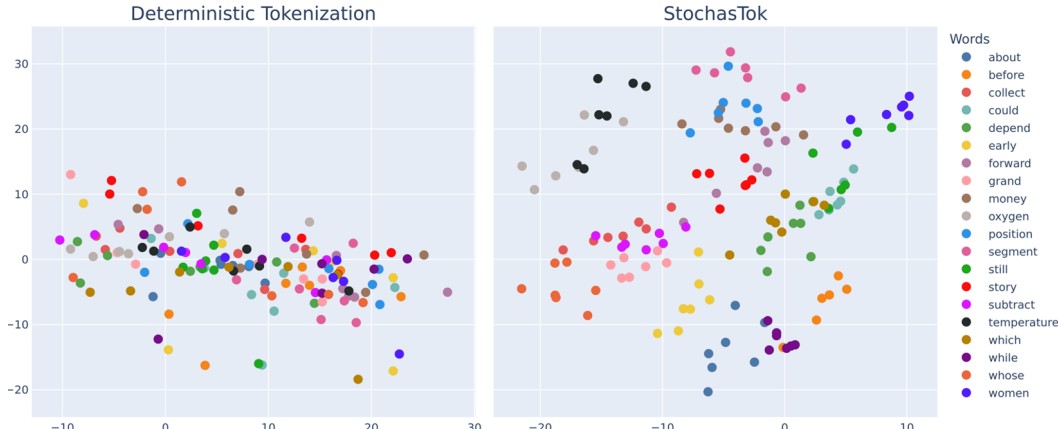

Figure 12: STOCHASTOK visibly results in the internal representations for alternative tokenizations of the same words being much more closely aligned.

Next, in Figure 12, we visualize the internal representations, both with and without stochastic tokenization. We fit a PCA model on the embeddings[3] of the top 1k most common words and then plot the results for alternative tokenizations of the same words, using a random sample of 20 words. We observe that, when using stochastic tokenization, the embeddings for alternative tokenizations of the same word are significantly more closely aligned and visibly capture subword-level structure.

For a more quantitative measure of this, in Figure 13, we plot how the mean distance between representations of alternative tokenizations of the same word evolves through the transformer layers. We observe that when trained with STOCHASTOK, each layer maps alternative tokenizations progressively closer to the same representation, while the deterministically pretrained model does not have this behavior.

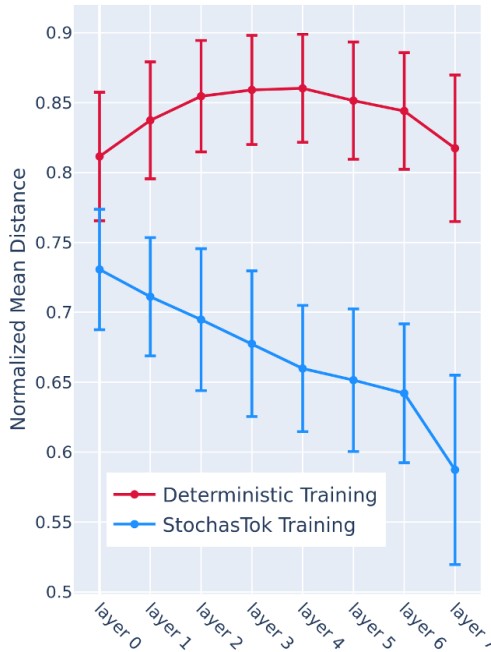

Figure 13: STOCHASTOK-trained models progressively map equivalent tokenizations closer together.

## 8 RELATED WORK

**Subtoken-level understanding.** Numerous papers have studied LLMs' surprisingly poor ability on subword-level tasks (Xu & Ma, 2024; Fu et al., 2024; Zhang et al., 2024; Shin & Kaneko, 2024; Edman et al., 2024; Marjieh et al., 2025; Kaushal & Mahowald, 2022; Itzhak & Levy, 2021, *inter alia*). However, solving these tasks remains challenging, despite improvements to core capabilities and reasoning in other measured benchmarks.

**Stochastic Tokenization.** Stochastic variants have been proposed for many tokenizers, including BPE-dropout for BPE (see Section 2), MaxMatch-dropout (Hiraoka, 2022) for WordPiece (Schuster & Nakajima, 2012), LCP-dropout (Nonaka et al., 2022) for LCP (Cormode & Muthukrishnan, 2002), and Subword Regularization and STM (Hiraoka et al., 2019) for Unigram (see Section 2). These prior methods are all tokenizer-specific, for example MaxMatch-dropout randomly omits the longest

---

[3]The activations after the final attention layer at the position of the last token for each word.

next subword when tokenizing with WordPiece, while LCP-dropout adds stochasticity by randomly partitioning the input before applying LCP tokenization. Similarly, Subword Regularization and STM rely on Unigram's unigram model for calculating tokenization probabilities using the FFBS or Viterbi algorithms (Scott, 2002; Viterbi, 1967), (but rather than choosing the highest probability tokenization, they instead sample from this distribution). Therefore, since almost all current LLMs use BPE tokenization, these methods are almost never applicable.

BPE-dropout is, therefore, the relevant baseline. As described in Section 3, compared to BPE-dropout, STOCHASTOK has several practical advantages: Firstly, to apply BPE-dropout, we require access to the exact merge hierarchy of the BPE tokenizer. By contrast, STOCHASTOK can be easily applied to any base tokenizer without any knowledge of the base tokenizer itself (it only requires knowledge of the model's vocabulary—a property of the model). Secondly, STOCHASTOK can be applied at any stage of the LLM pipeline, even to pretrained models, since it preserves the same vocabulary as the original tokenizer. In contrast, switching between BPE and BPE-dropout changes the possible vocabulary, leading either to out-of-vocabulary tokens or requiring a change to the model. Finally, STOCHASTOK is essentially a lightweight processing step after tokenization, meaning it can be used in conjunction with fast, compiled implementations of base tokenizers. By contrast, BPE-dropout requires tokenizing from scratch and compiled implementations of BPE-dropout for predefined BPE tokenizers (i.e., a pre-specified vocabulary and merge hierarchy) are not readily available, thus often making BPE-dropout prohibitively expensive.

**Byte-level models.** An alternative line of work in improving character-level understanding is byte-level or 'tokenizer-free' models, which operate directly on characters. This approach removes the inductive bias imposed by tokenizers' vocabularies and naturally handles unusual words and typos. However, the naïve approach is prohibitively inefficient due to increased sequence lengths. As a result, approaches such as hierarchical architectures, local convolutions, patching mechanisms, or auxiliary losses, are necessary to bring down the effective sequence lengths (Al-Rfou et al., 2019; Clark et al., 2022; Yu et al., 2023; Pagnoni et al., 2024). However, these come at the cost of added complexity and still substantially higher computational requirements (Xue et al., 2022; Nawrot et al., 2022). Consequently, tokenization-based models currently remain more compute-efficient, and more practical in general. With STOCHASTOK we enable models to get the benefits of byte-level understanding without needing to move to an alternate framework.

## 9    DISCUSSION AND FUTURE WORK

While there are adoption costs with any changes to the LLM pipeline, STOCHASTOK minimizes these through its simplicity, wide compatibility, and demonstrated ability to be applied to existing pretrained models. Looking ahead, a valuable addition would be to apply STOCHASTOK 's on a larger scale to investigate other potential benefits, such as greater robustness to spelling mistakes and other general improvements. In this paper, we focus only on English, and it would also be interesting to explore the effect of STOCHASTOK on languages with different alphabets, structure, and levels of morphology. Finally, combining STOCHASTOK with recent orthogonal advances in tokenization, such as Liu et al. (2025), represents another promising direction for future research.

## 10    CONCLUSION

Our experiments demonstrate that incorporating STOCHASTOK at any stage of training dramatically enhances language models' ability to represent subword-level structures central to human language perception. Tokenization has recently received less attention than other methods, such as finetuning and prompting techniques, since its position at the start of the pretraining pipeline often makes experimentation prohibitively expensive. Our work shows that tokenization modifications can be exceptionally effective, not only at the pre-training stage but also in the continued pre-training and post-training stages. Our efficient, cheap changes can help fix pervasive idiosyncrasies and lead to significant improvements in language understanding. Given the stark performance benefits demonstrated here, we are excited to assess the impact of STOCHASTOK on more challenging tasks such as coding, algebra, or scientific reasoning when applied to more capable models. We hope our work encourages renewed exploration of tokenization schemes to bridge the gap between human and machine language perception.

## ACKNOWLEDGMENTS

We thank the contributors of OpenWebText and the maintainers of SuperTinyLanguageModels for making their resources publicly available under the MIT License. AS is supported by the EPSRC Centre for Doctoral Training in Modern Statistics and Statistical Machine Learning (EP/S023151/1). YWT's research is supported by the Ministry of Digital Development and Information (MDDI) under the Singapore Global AI Visiting Professorship Program (Award No. AIVP-2024-002).

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

# SUPPLEMENTARY MATERIAL

## TABLE OF CONTENTS

# A  TOKENIZERS

## A.1  BPE TOKENIZATION

**Construction**
The tokenizer is constructed by initializing the vocabulary as individual characters and then iteratively adding the most frequent adjacent token pair in the 'training dataset' until the desired vocabulary size is reached. This yields a vocabulary and a hierarchy of merge rules.

**Encoding**
The dataset is initially tokenized as individual characters. Pairs of tokens are then merged according to the hierarchy of merge rules until there are no more merges available.[4]

**Decoding**
The text strings corresponding to each token ID are simply looked up and joined together.

## A.2  UNIGRAM TOKENIZATION

**Construction**
In contrast to BPE, Unigram starts with a large candidate vocabulary of possible subword units and removes elements to get down to the desired vocabulary size. Tokens are removed from the vocabulary by modeling the dataset as a Unigram model and removing the token that results in the smallest increase in log-likelihood of the dataset considering all possible tokenizations. This relies on using the Viterbi algorithm to compute probabilities of all possible tokenizations. It also relies on using the Expectation-Maximization (EM) to optimize the vocabulary and the probability of the dataset simultaneously. The result is a vocabulary and corresponding probabilities of each token (i.e., a Unigram model of the dataset).

**Encoding**
All possible tokenizations are considered, and the one with the highest probability under the unigram model is chosen. This involves using the Viterbi algorithm to find the highest probability tokenization.

**Decoding**
Same as BPE: The text strings corresponding to each token ID are simply looked up and joined together.

## A.3  STOCHASTOK TOKENIZATION - PSEUDOCODE

---
**Algorithm 1** STOCHASTOK: Construction of `splits`

---
1: **Require:** Tokenizer (e.g. `tiktoken`'s GPT-2 tokenizer)
2: $\mathcal{V} \leftarrow$ Tokenizer vocabulary
3: `splits` $\leftarrow \{\}$                                             Initialize an empty dictionary
4: **for each** token $s$ in $\mathcal{V}$ **do**
5:     $t \leftarrow \text{encode}(s)$                                          Get the token id
6:     `splits`$[t] \leftarrow [\,]$                              Initialize empty list for this token
7:     **for each** possible split index $i$ from 1 to $\text{len}(s) - 1$ **do**
8:         $s_1, s_2 \leftarrow s[:i], s[i:]$                        Split string $s$ into two substrings
9:         **if** $s_1$ and $s_2$ in $\mathcal{V}$ **then**
10:             $t_1, t_2 \leftarrow \text{encode}(s_1), \text{encode}(s_1)$              If both substrings are in the vocab
11:             `splits`[t].append($(t_1, t_2)$)                         Add this possible split
12:         **end if**
13:     **end for**
14: **end for**

---

---
[4]WordPiece (Schuster & Nakajima, 2012) can be seen as a variant of BPE with merges during encoding chosen by token length rather than the original merge rules.

---

**Algorithm 2** STOCHASTOK: Tokenization

---

1: **Require:** Tokenizer
2: **Require:** `text`: The input text to tokenize
3: **Require:** `splits`: Dictionary of possible splits for each token
4: **Require:** `expand_prop`: Expansion proportion (e.g. = 0.01)
5: `tokenized` ← Tokenizer(`text`)                                 Apply standard tokenization
6: `num_to_expand` ← len(`tokenized`) ∗ `expand_prop`
7: **for** _ in $1 \cdots$ `num_to_expand` **do**
8:   $i \leftarrow$ randomInteger$(1, \text{len}(\texttt{tokenized}))$                Choose a random position
9:   $t \leftarrow \texttt{tokenized}[i]$
10:   **if** $t$ in `splits` **and** `splits`$[t]$ not empty **then**
11:     $(t_1, t_2) \leftarrow$ randomChoice(`splits`$[t]$)          Replace with a random split
12:     $\texttt{tokenized} \leftarrow \texttt{tokenized}[1 : i-1] + [t_1, t_2] + \texttt{tokenized}[i+1 :]$
13:   **end if**
14: **end for**
15: **return:** `tokenized`

---

## A.4 STOCHASTOK TOKENIZATION - ANOTHER ILLUSTRATIVE EXAMPLE

Example vocabulary of base tokenizer:

```
vocabulary = [_, h, u, g, b, m, hu, ug, hug, bug]
```

Build `token_splits` which, for each token, contains a list of all possible pairs of component tokens that are themselves in the vocabulary.

```
token_splits = {
        ug:[(u,g)],
        hu:[(h,u)],
        hug:[(h,ug),(hu,g)],
        bug:[(b,ug)],
        ugs:[(ug,s)]
}
```

Examples of possible expansions:

```
original:  [hug] → all possible expansions:  [hu g], [h ug], [h u g]

original:  [bug] → all possible expansions:  [b ug], [b u g]

original:  [m ug] → all possible expansions:  [m u g]
```

# B LANGUAGE GAME AND MATH DATASETS

In this section, we provide details of each of the three evaluation datasets: LangGame, CUTE, and multi-digit addition.

## B.1 LANGGAME

We create a new benchmark, 'LangGame,' to test subword-level understanding in LLMs. LangGame is a multiple-choice based dataset, allowing for easy evaluation, and it is suitable for small models. Here, we describe its construction in detail. The language game consists of six types of questions:

1. *Which word has the most letter '#'s?*
2. *Which word contains '#'s?*
3. *Which word starts with '#'s?*
4. *Which word ends with '#'s?*
5. *Which word is longest?*
6. *Which word is shortest?*

We include multiple phrasings for each type of question by constructing the question with a template and randomly replacing the placeholders.

Question template:

```
"<WHICH><WORD> <question>?  <THE><OPTIONS><ARE>:  <options>.  Answer:
    <answer>."
```

Synonyms for placeholders:

```
<WHICH>:  ["Which", "What"]
<WORD>:  [" word", "", " string", " option", " choice", " option word",
    " option string"]
<THE>:  ["The", "The possible", "The available"]
<OPTIONS>:  [" options", " choices", " option words", " option strings"]
<ARE>:  [" are", ""]
```

This results in $2 \times 7 \times 3 \times 4 \times 2 = 336$ possible phrasings for each question.

Question strings are then chosen from:

```
"has the most letter '<AUX>'s?",
"contains '<AUX>'",
"starts with '<AUX>'",
"ends with '<AUX>'",
"is the longest",
"is the shortest",
```

Option words and answers are sampled randomly from the top 1k English words, and sub-strings for the `"contains"`, `"starts with"`, and `"ends with"` question types are sampled randomly from the answer with length $\geq 1$ and $\leq$ the answer length, and we generate 10k train and 1k validation examples. For the experiments in Figure 6, for the train and validation sets, substring lengths are $\geq$ half the answer word length, and for the holdout set, substring lengths are $<$ half the answer word length. An example of each type of question is given in Table 1.

We evaluate accuracy based on whether the probability of the correct option is the highest compared to all the alternative options in the question, but additionally when looking at generations, we find that the STOCHASTOK-finetuned models generate the correct answer over all other possible next tokens.

## B.2 CUTE BENCHMARK

We also evaluate on the Character-level Understanding of Tokens Evaluation (CUTE) benchmark (Edman et al., 2024). CUTE contains 14 question types:

| | Task | Question | Answer |
|---|---|---|---|
| 1 | Spelling | Spell out the word: there | t h e r e |
| 2 | Inverse Spelling | Write the word that is spelled out (no spaces): t h e r e | there |
| 3 | Contains Char | Is there a 'c' in 'there'? | No |
| 4 | Contains Word | Is there 'the' in 'the sky is blue'? | Yes |
| 5 | Orthographic | Closer in Levenshtein distance to 'happy': glad or apply? | apply |
| 6 | Semantic | More semantically related to 'happy': glad or apply? | glad |
| 7 | Char Insertion | Add 'b' after every 'e' in 'there' | thebreb |
| 8 | Word Insertion | Add 'is' after every 'the' in 'the sky is blue' | the is sky is blue |
| 9 | Char Deletion | Delete every 'e' in 'there' | thr |
| 10 | Word Deletion | Delete every 'the' in 'the sky is blue' | sky is blue |
| 11 | Char Substitution | Replace every 'e' with 'a' in 'there' | thara |
| 12 | Word Substitution | Replace every 'the' with 'is' in 'the sky is blue' | is sky is blue |
| 13 | Char Swapping | Swap 't' and 'r' in 'there' | rhete |
| 14 | Word Swapping | Swap 'the' and 'is' in 'the sky is blue' | is sky the blue |

Table 2: Examples of the CUTE benchmark of language composition, similarity, and manipulation tasks.

We use the eight subword-level question types (types 1, 2, 3, 5, 7, 9, 11, and 13). The original benchmark was designed for zero-shot evaluation of full-scale industrial models, and hence, it only includes a test set. To evaluate our smaller pre-instruction finetuning models, we require additional training examples for finetuning, hence we generate more questions for each of the eight types. We generate questions by randomly sampling words from the top 1k English words. Consistent with the multiple-choice format of the open-source baseline code (Hillier et al., 2024), we also create incorrect answer options. For questions where the answer is an option in the question (question types 3 and 5), the incorrect options are the other options in the question (e.g., Yes/No). For questions where the answer is a word (question type 2), the incorrect options are other randomly sampled words from the other top 1k English words. Finally, for the remaining question types where the answer is a sequence of letters (question types 1, 7, 8, 11, 13), the incorrect options are generated by substituting and reordering letters in the correct answer. Results on each of the individual CUTE tasks over training are shown in Figure 14.

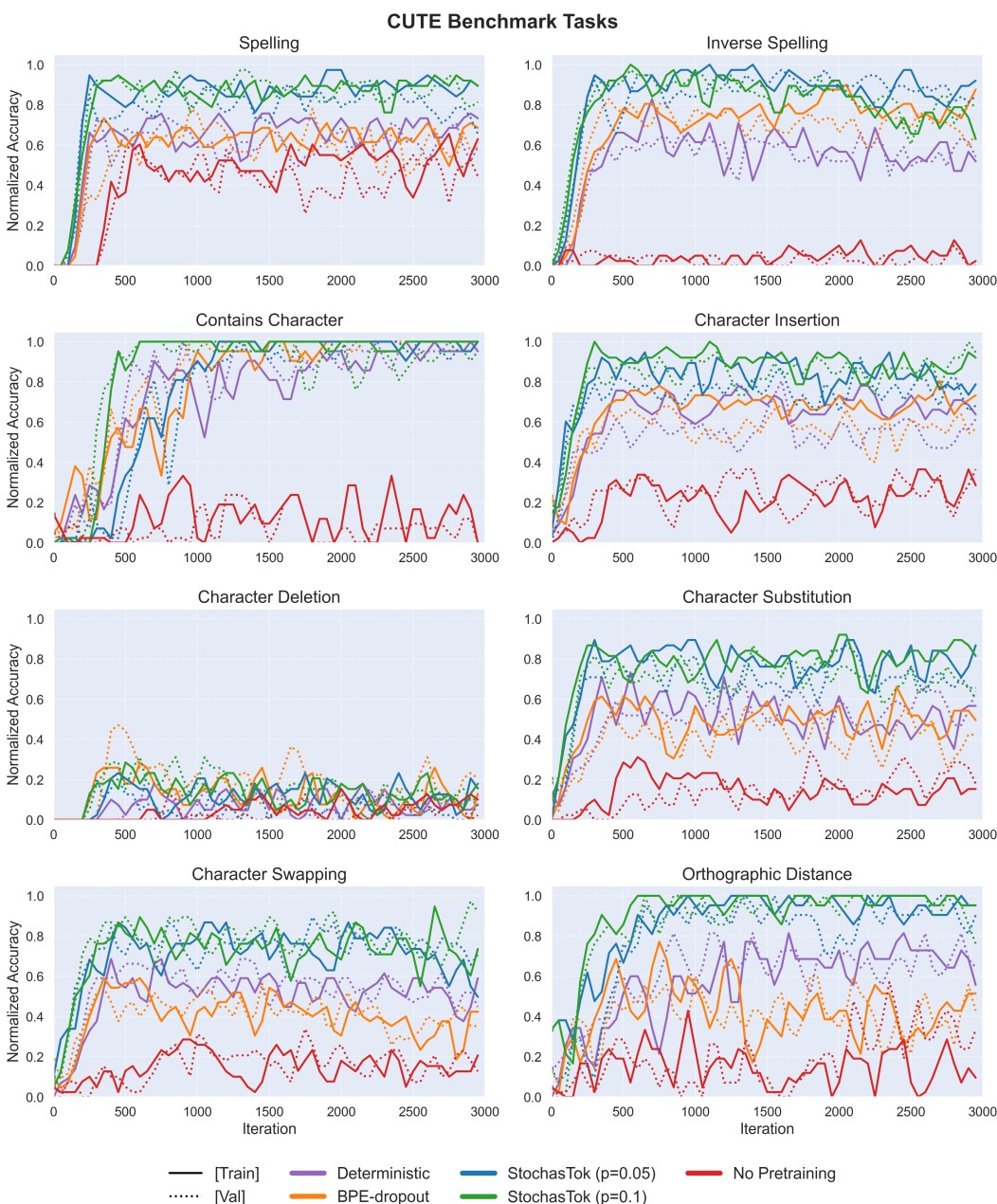

Figure 14: Performance on each of the tasks within the CUTE benchmark over training. (Accuracy normalized so that random guessing is zero.)

## B.3 MULTI-DIGIT ADDITION

For the multi-digit addition experiments, we sampled pairs of integers up to 1000. The answer is reversed as per the procedure in Lee et al. (2023), and we then train on a stream of examples, e.g., `'$ 151+687=838 $ 328+869=7911 $ 752+917=9661 $ 747+303=0501 $ 857+579=6341 $ ...'` with the setup described in Section C.1.

## C TRAINING SETUPS

In this section, we provide full details of the training setups used in the paper. For STOCHASTOK's hyperparameter $p$, we find that careful tuning is not required and that any value between 0.01 and 0.2 gives good performance. Throughout the paper, we show results with $p = 0.1$ (and also include $p = 0.05$ in some places as effectively an extra seed). For BPE-dropout, we use $p = 0.1$ as suggested in the original paper.

### C.1 50M PARAMETER MODEL SETUP

We build on the baseline 50M-parameter model setup in the open-source SuperTinyLanguageModels repo (Hillier et al., 2024), which is trained on the OpenWebText dataset (Gokaslan & Cohen, 2019) and uses the GPT-2 BPE tokenizer from the `tiktoken`[5] library. The pretraining benchmarks evaluated on (see Figure 3) are ARC (Clark et al., 2018), Blimp (Warstadt et al., 2020), HellaSwag (Zellers et al., 2019), Winograd (Sakaguchi et al., 2021). The full set of hyperparameters for pretraining are given in Table 3.

| Model | |
|---|---|
| number of layers | 8 |
| ffn type | SwiGLU |
| ffn dimension | 1320 |
| number of attention heads | 16 |
| group size | 4 |
| hidden dim | 512 |
| tokenizer type | gpt2 |
| vocab_size | 50257 |
| max context window | 512 |
| positional_encoding_type | RoPE |
| Training | |
| batch_size | 480 |
| total iterations | 30000 |
| warmup iterations | 5000 |
| dropout | 0.1 |
| Optimizer | |
| optimizer | AdamW |
| lr | 6.0e-04 |
| min_lr | 6.0e-05 |
| lr_scheduler | Cosine |
| weight_decay | 0.1 |

Table 3: The baseline setup as in Hillier et al. (2024)—a 50M-parameter transformer LLM.

For fine-tuning (as in Figure 4), we train for a further 3k iterations with a learning rate of 1.0e-04 on the LangGame or CUTE datasets. For continued pretraining (as in Figure 9) we similarly train for a further 3k iterations with learning rate 1.0e-04 on OpenWebText.

### C.2 275M PARAMETER MODEL SETUP

For the 275M parameter model, we follow Jordan et al. (2024a), training on FineWeb (Penedo et al., 2024) with the hyperparameter setup given in Table 4.

### C.3 GPT-2 CONTINUED PRETRAINING SETUP

We initialize the model from the publicly available pretrained weights and architecture on Huggingface at https://huggingface.co/openai-community/gpt2. For the continued pretraining, we train for 7k steps with a constant learning rate of $1.0e-4$ and a batch size of 128. For the finetuning on LangGame tasks presented in Figure 10, we finetune for 2k steps, again with a constant learning rate of $1.0e-3$ and a batch size of 512.

---

[5] github.com/openai/tiktoken

| Model | |
| --- | --- |
| number of layers | 12 |
| ffn type | ReLU |
| ffn dimension | 768 |
| number of attention heads | 6 |
| head dimension | 128 |
| tokenizer type | gpt2 |
| vocab_size | 50257 |
| max context window | 1024 |
| positional_encoding_type | RoPE |
| Training | |
| batch size | 384 |
| total iterations | 60000 |
| cooldown frac | 0.4 |
| Optimizer | |
| weights optimizer | Muon |
| head, embeddings, biases optimizer | AdamW |
| head lr | 0.044 |
| embeddings lr | 0.12 |
| biases lr | 0.008 |
| weights lr | 0.01 |

Table 4: The baseline setup as in Jordan et al. (2024a)—a 275M-parameter transformer LLM. The changes made to the baseline are training for 60k iterations (as opposed to the 1770 iterations of the original baseline, since the baseline config was set up as a demo) and reducing all the learning rates by a factor of 5 (needed to stabilize training of all models when training for longer).

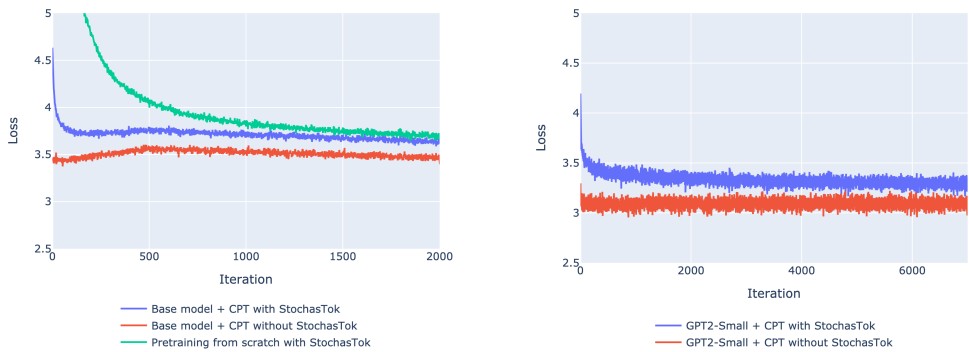

Figure 15: Training loss on OpenWebText during continued pretraining for the 50M STLM base model and GPT-2.

# D    ANALYSIS DETAILS

In the following section, we provide additional details and results of the visualizations in Section 7.

## D.1    DIFFERENT PROMPT COMPLETIONS SETUP

Further examples of completions from multiple different tokenizations of the same prompts are given in Figure 16. The prompts are generated by GPT-4o. We find that the deterministic tokenization-trained model is very sensitive to prompt tokenization and quickly breaks down when given alternative tokenizations of the same prompt. By contrast, the STOCHASTOK-trained model is much more robust to prompt tokenization.

## D.2    EMBEDDING VISUALIZATION SETUP

As described in the main text, the activations for a word are taken as the residual stream activations after the final transformer layer. If the word is tokenized into multiple tokens, we use the position of the final token. We use the standard procedure of normalizing to zero mean and unit standard deviation before fitting the PCA model.

## D.3    DISTANCE OVER LAYERS VISUALIZATION SETUP

In Figure 13, we plot the mean distance between embeddings of different tokenizations of the same word over the layers of the model. For normalization to allow comparison between different models, we first normalize all embeddings to have unit length. We then evaluate the average distance between embeddings for pairs of different words in the model, and we divide by this average distance metric.

| Different prompt tokenizations | Deterministic Training | StochasTok Training |
|---|---|---|
| Deep in the forest , they found | a small , small , and | a small , white - tailed |
| Deep in the forest , they fou nd | own , and the water is | a small , white - tailed |
| De ep in the forest , th ey fo un d | , th ey fo un d | a small group of people who |
| D e ep in the forest , t hey found | a small , small , and | a small , white - tailed |
| Deep in the forest , they fou nd | own , and the water is | a small , white - tailed |
| De ep in the forest , th e y found | the first evidence of the presence | a small , white - tailed |
| She stepped onto the stage , and the | crowd was cheering . | crowd was silent . |
| She step ped onto the stage , and th e | ep , the crowd was cheering | crowd was silent . |
| S he stepped onto th e stage , and th e | stage . " I | crowd was silent . |
| She stepped on to the stage , a nd the | first time I saw him . | crowd was silent . |
| She ste pped onto the stage , a nd th e | - t - t - t | crowd was silent . |
| S he stepped onto the stage , and the | sound of the sound of the | crowd was silent . |
| In the picture , the waves crashed against the | wall of the building , and | wall of the building , and |
| In the picture , the wave s cr ashed again st t he | the wind , and the wind | wall , and the wind blew |
| In the picture , the waves cr ashed again st th e | - t ung s . | wall , and the wind blew |
| In th e picture , the waves cr ashed again st t he | the waves . The | wall , and the ground was |
| I n the picture , the waves crashed against the | wall . " I | wall , and the ground shook |
| In the picture , th e waves crashed again st t he | , and the wind blew out | wall , and the ground shook |
| The scientist carefully adjusted the microscope , searching for | the source of the light . | the most common and most common |
| The scientist car efully adjusted the microscope , searching for | the source of the data . | the most common and most common |
| The scient ist carefully adjusted t he microscope , se arching f or | so a o - d ino | the most common and most common |
| The scient ist car efully adjusted the microscope , se arching for | the first time in a decade | the most common and most common |
| The scient ist car efully adjusted th e microsc ope , searching for | the source of the gas . | the most common and most common |
| The scient ist car efully adjusted the microsc ope , se arching f or | so - shaped , and the | the most common cause of death |
| The journalist pressed record , ready to capture the | truth . ◆ ◆ | truth . The story |
| Th e journalist pressed record , ready to capture t h e | ◆ ◆ s true identity . | truth . The story |
| The journalist pres sed record , ready to capt ure the | world . ◆ ◆ | truth . The story |
| T he journal ist pres sed record , ready to capture the | essence of the universe . | essence of the story . |
| The journalist pressed record , ready to capt ure th e | - fi into the world of | truth . The story |
| T he journalist pressed record , ready to capture t he | 's face . " | truth . The story |

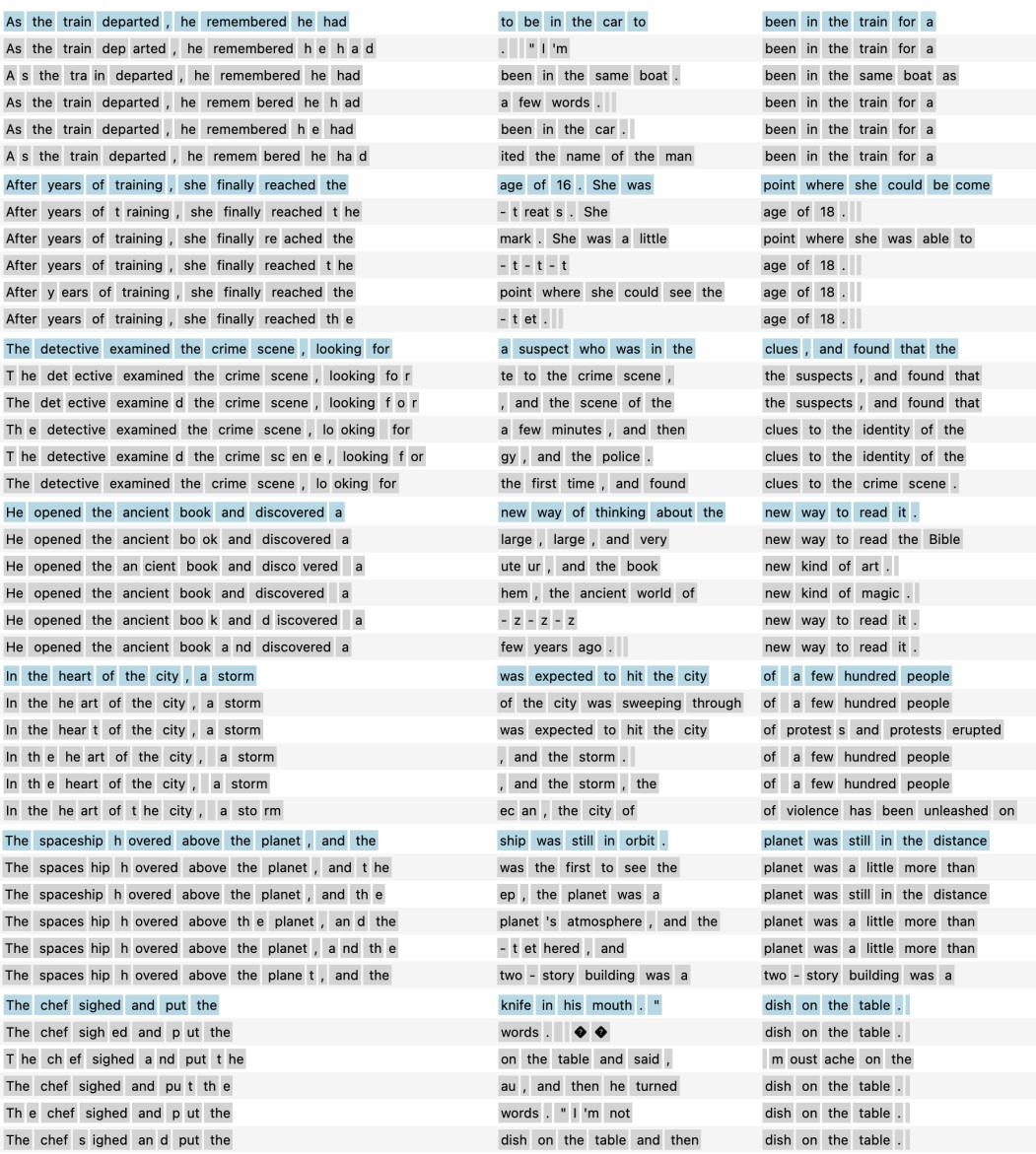

Figure 16: Example responses with different tokenizations.

