# OpenReview forum: "StochasTok: Improving Fine-Grained Subword Understanding in LLMs"
_ICLR.cc/2026/Conference — ICLR 2026 Poster_

### Official Review · Reviewer_ZDYH · 2025-10-17

**Soundness:** 3
**Presentation:** 3
**Contribution:** 3
**Rating:** 8
**Confidence:** 3

**Summary:**

The paper proposes STOCHASTOK, a simple stochastic tokenization scheme that preserves the original vocabulary and introduces *training-time only* reversible splits of existing tokens to expose subword/character structure. The approach aims to improve tokenization invariance without changing model architecture or deployment-time tokenization. Empirically, it delivers consistent gains on curated subword “language games” and cross-tokenizer multi-digit addition while keeping standard LM metrics roughly unchanged on small/medium models; qualitative analyses (e.g., embedding alignment across segmentations) support the intuition. The idea is elegant and easy to integrate (especially for CPT), but evidence on larger models and real-world tasks is currently limited.

**Strengths:**

1. **Simplicity & compatibility**: Fixed vocabulary, tokenizer-agnostic, and *train-time only* noise makes it a low-friction drop-in (particularly for CPT) with minimal code churn.
2. **Clear signal on subword skills**: Reproducible improvements on subword-aware tasks and cross-tokenizer arithmetic transfer, with no obvious small-scale regressions on general LM metrics.
3. **Practical deployment story**: Inference remains deterministic, avoiding train–test segmentation mismatch common to methods that alter the tokenizer itself.
4. **Plausible mechanism**: Training with multiple valid segmentations encourages segmentation-invariant internal features; analyses suggest layer-wise convergence toward shared representations.

**Weaknesses:**

1. **External validity at scale**: No results on ≥4B/7B models; claims of robustness and easy integration would be stronger with short-budget CPT evidence at those scales.
2. **Evaluation scope**: “Real” math (e.g., GSM8K/MATH) and broader tasks (MMLU/BBH/code/RC) are missing; current gains mostly establish tokenization invariance rather than end-task improvements.
3. **Compute/cost parity**: No end-to-end cost curves (length inflation, throughput, VRAM, wall-time) or budget-matched comparisons, making Pareto efficiency unclear.

**Questions:**

1. **Scale-up sanity check**: Can you run short-budget CPT on one **4B** and one **7B** model (e.g., 3–10k steps, \(p \in \{0.05, 0.1\}\)) and report subword tasks, a small **MMLU** slice (no-regression), and **throughput/length/VRAM**?
2. **“Real” math**: On **GSM8K** (dev is fine), compare deterministic vs **BPE-dropout** vs **STOCHASTOK** under **equal compute**, including training curves and final accuracy; also test cross-tokenizer transfer at this scale.

---

> ### Author Response · Authors · 2025-11-21
>
> We thank the reviewer for their high score (8) and encouraging feedback, e.g the method being “elegant and easy to integrate". Moreover, we are extremely grateful for the thorough technical review and valuable feedback which will help us to improve the paper.
>
> **W1/Q1. Scale up CPT experiments:** Thank you for suggesting this—we agree this would be a valuable addition. We are a small academic lab with limited compute, but we will endeavor to have this for the final camera ready version of our paper.
>
> **W2/Q2. “Real” math tasks:** Thank you again for this suggestion. We agree that math tasks like GSM8K are the next logical step. These tasks however, rely heavily on reasoning, not just arithmetic calculation. We expect our base models (50M/275M) to currently be too small to coherently reason through these tasks sufficiently, hence we focused on multi-digit addition–a task that requires sub-token understanding without requiring reasoning. Given our results in Figure 8 (where stochastok enables the model to grok multi-digit addition) we are optimistic that stochastok could also improve general manipulation of multi-digit numbers and hence improve GSM8K performance when used with larger models with reasoning capacity. We are currently sourcing compute to allow us to run larger pretraining experiments, with the aim to then test on more realistic tasks like GSM8K as suggested. In the meantime, we hope this paper will inspire future work from industrial labs to test it out at larger scales.
>
> **Q2.2/W3 Compare under equal compute:** To clarify, the compute budget was kept fixed between tokenizers during training since all models were trained on the same number of tokens. This meant that the models trained with stochastok and BPE-dropout consequently saw slightly less text than the model trained with the standard tokenizer. Hence stochastok has significantly higher pareto efficiency as it boosts performance at the same compute budget. In addition, particularly on the addition task, stochastok allowed the model to learn/grokk the task much more quickly which significantly reduced the time-to-convergence/compute cost.
>
> Furthermore, we found that the *inference-time* cost increase was almost zero when using stochastok. This is because, all post-training (e.g see Figure 1) was done with standard tokenization, and we found that the final model therefore tended to output responses with standard (ie. shortest) tokenization.
>
> Thank you for asking about this. We agree that these are important and interesting details and we will add plots and discussion of throughput, VRAM, wall-time, train/inference costs, etc, to the paper.
>
> ---
>
> We thank the reviewer for their careful review and genuinely helpful suggestions. We hope we have answered your questions sufficiently. Please let us know if you have any more questions or comments as we would be happy to discuss further.

---

> > ### Comment · Reviewer_ZDYH · 2025-11-28
> >
> > Thanks for the reply. It is a good paper. I will maintain my score.

---

> > > ### Author Response · Authors · 2025-11-29
> > >
> > > Thank you for replying to our rebuttal and for continuing to advocate for our paper. We really appreciate your support.

---

### Official Review · Reviewer_eKHe · 2025-10-24

**Soundness:** 3
**Presentation:** 3
**Contribution:** 3
**Rating:** 6
**Confidence:** 4

**Summary:**

The paper proposes a new stochastic tokenization approach called StochasTok. StochasTok allows for flexible, alternative tokenizations of the same token. Results on various datasets including language games, pre-training small-scale models (50-275M parameters) offer large performance gains.

**Strengths:**

* The paper is well written and easy to read.
* Tokenization is a really important and often neglected area of LMs.
* The method proposed is simple and works well
* The paper offers a comprehensive analysis and discussion of experiments and results.

**Weaknesses:**

* Really minor (without trying to be "Reviewer 2"): the models tested are really small for 2025 standards. There is a risk that the gains might not generalize to larger scales.

**Questions:**

- Perhaps experimenting with slightly larger models might offer a clearer picture if the gains from your method are similar in larger LMs (e.g. ~1B).

---

> ### Author Response · Authors · 2025-11-21
>
> We thank the reviewer for the positive score and feedback, eg., that our paper includes “comprehensive analysis and discussion”. We are glad that the reviewer recognizes that tokenization is an often neglected but critical area.
>
> **1. Model scale:** We agree that the models tested are small for 2025 standards. However, as an academic lab we are unable to run 1B+ size experiments as pretraining is surprisingly expensive - (As an example, [1] states that running training for their 1.7B SmolLM2 model costs *“around 1e23 FLOPs, or $250,000 USD worth of GPU compute”*). Instead, we designed our experiments to show trends, demonstrating that our results hold when scaled between two model sizes. We hope our paper will inspire industrial labs with access to more compute to try out stochastok, or to develop other related alternatives to standard tokenization. We agree that, if possible, 1B+ experiments would be valuable to the paper. We are currently trying to source compute to run these, and hope to get these for the camera ready version of the paper.
>
> [1] - SmolLM2: When Smol Goes Big -- Data-Centric Training of a Small Language Model, L B Alla, et al. 2025 (https://arxiv.org/abs/2502.02737v1)
>
> —
>
> We thank the reviewer for their encouraging feedback. We hope we have been able to address your concern about scale. Reviewer dC4j raised the same comment about scale and gave a Score=8, noting that, given the complexities of such experiments for academic labs, they did not expect larger experiments. We hope this reviewer will also appreciate this. Please let us know if you have any more questions or concerns as we would be more than happy to discuss further. If we have been able to address all your concerns we humbly ask if you would consider raising your score.

---

### Official Review · Reviewer_HeMs · 2025-10-31

**Soundness:** 3
**Presentation:** 3
**Contribution:** 3
**Rating:** 6
**Confidence:** 3

**Summary:**

This paper introduces StochasTok, a novel stochastic tokenization method designed to address a key limitation of large language models: their poor performance on subword-level reasoning tasks. Standard tokenizers treat words as opaque symbols, obscuring the internal character structure and making tasks like counting letters or performing multi-digit arithmetic surprisingly difficult for even state-of-the-art models.

The proposed method operates as a lightweight post-processing step that randomly splits tokens into smaller, valid subtokens from the existing vocabulary, thereby exposing the model to the morphological composition of words during training. The authors demonstrate through extensive experimentation that StochasTok significantly enhances subword reasoning capabilities. Their approach consistently and substantially outperforms strong baselines in both pretraining and finetuning scenarios, achieving near-perfect accuracy on language game benchmarks like LangGame and CUTE, and enabling models to rapidly learn complex tasks such as multi-digit addition.

**Strengths:**

1. The proposed StochasTok approach offers an elegantly simple yet effective solution to the fundamental limitation of subword understanding in LLMs. Its implementation as either a pretraining enhancement or a lightweight finetuning step makes it highly practical and accessible.
2. The authors provide comprehensive validation across multiple domains - from language games (LangGame, CUTE) to mathematical reasoning (multi-digit addition) - demonstrating the method's versatility and robust performance gains.
3. The paper provides some insights into the method's internal mechanisms through embedding visualizations

**Weaknesses:**

See questions.

**Questions:**

1. I acknowledge the contributions of this work. However, from my understanding, StochasTok appears to be a special case of BPE-dropout? Is it true that for every tokenization generated by StochasTok, there exists an equivalent BPE-dropout tokenization? If so, could the authors provide an intuitive explanation for why StochasTok performs so much better than BPE-dropout in the experiments? Are there key differences in how the stochasticity is applied or how the model learns from these variations that lead to the significant performance gap? Could the authors provide a more detailed comparative analysis between the two?
2. The paper states that "In BPE, intermediate tokens not present in the final tokenized training dataset are removed from the vocabulary, meaning BPE-dropout can produce tokens outside the original vocabulary", which I'm not sure. If true, how does it encode user inputs during inference, given this vocabulary mismatch?
3. In Figure 1, the "no pretraining" model shows a rapid increase in both training and validation accuracy very early in training, followed by a sudden decrease. What is the authors' explanation for this phenomenon?
4. In Section 5, which focuses on multi-digit addition, what was the range of digits for the numbers used in the training and validation sets?
5. Could you please report performance on individual subtasks of both LangGame and CUTE (e.g., "Inverse Spelling", "Char Deletion")? This would provide clearer insights into the model's capabilities.

---

> ### Author Response · Authors · 2025-11-21
>
> We thank the reviewer for acknowledging the elegance and versatility of stochastok, and for their detailed analysis that will help us significantly improve the paper. We appreciate the opportunity to clarify the technical distinctions regarding BPE-dropout.
>
> **Q1. StochasTok vs. BPE-dropout:** Thank you for this question. Crucially, StochasTok is not a special case of BPE-dropout. Firstly, BPE-dropout will contain tokenizations that are not present with StochasTok since BPE-dropout may effectively split two adjacent tokens and then merge them (e.g. “token” “ization” -> “tok” “eni” “zation” ) which is not possible with stochastok. Instead Stochastok follows the intuition of allowing models to “see inside” tokens by just splitting them up with some small probability. Secondly, unlike stochastok, BPE-dropout can introduce additional unknown tokens (UNKs) (see response to Q2). Thirdly, stochastok and BPE-dropout will induce different token frequencies and patterns. Empirically we find that these differences from BPE-dropout make stochastok significantly stronger. This is an interesting question and we will update the paper to include this discussion.
>
> **Q2. How does BPE/BPE-dropout deal with OOV tokens?:** In BPE, if the tokenizer merges generate an OOV subword (very rare but possible), the tokenizer falls back to an unknown token (UNK). In BPE-dropout, the likelihood of generating an OOV subword is significantly higher. As a result the BPE-dropout paper proposes two ways to deal with this: *“There are two ways of building vocabulary for models trained with BPE-dropout: (1) take the vocabulary built by BPE; then the segmented with BPE-dropout text will contain a small number of unknown tokens (UNKs); (2) add to the BPE vocabulary all tokens which can appear when segmenting with BPE-dropout.”* Both of these are not ideal, since option (1) adds more UNKs, and option (2) requires changing the vocabulary of the model which is a problem for adapting pretrained models. By contrast stochastok splits tokens only into other tokens in the vocab, meaning it does not change the vocab and does not add additional UNKs. Thank you, this is an important detail and we will add it to the paper.
>
> **Q3. Shape of "No pretraining" curve (Figure 1):** We thank the reviewer for this interesting question. Since the model is trained from scratch with no pretraining, it lacks the general knowledge to tackle this task so we are likely in an overfitting regime. Our intuition is that the model could be initially memorizing each example, before it reaches the capacity of the network, at which point it then fails to generalize and properly fit the new examples as well as forgetting the old questions.
>
> **Q4. Math details:** As described in Appendix B.3, we sampled pairs of integers up to 1000, with the task then involving adding these numbers (e.g., $151 + 687$). We will add this into the main text for clarity.
>
> **Q5. Breakdown of CUTE Subtasks:** Thank you for this suggestion. We agree the per-task breakdown of the CUTE benchmark would be interesting to include in the paper, and we have added it to the updated version - **See new Figure 14**. We find that stochastok significantly improves learning on all the tasks, with the exception of “character deletion”, which we observe to be difficult for all the tokenizers.
>
> ---
>
> We thank the reviewer again for their careful review of our submission. The question comparing to BPE-dropout was particularly insightful. We hope we have sufficiently addressed all of your questions. Please let us know if you have any remaining concerns. If we have sufficiently addressed them all we humbly ask if you would consider raising your score.

---

### Official Review · Reviewer_dC4j · 2025-11-02

**Soundness:** 3
**Presentation:** 3
**Contribution:** 3
**Rating:** 8
**Confidence:** 3

**Summary:**

The paper introduces a new tokenization method STOCHASTOK to improve subword-level understanding of language models. The method is simple, efficient, and compatible to existing tokenization methods. Experiments show that pretraining models with STOCHASTOK improves model perofrmance on language game tasks and multi-digit addition. The paper also demonstrates that applying STOCHASTOK during fine-tuning can enhance the subword understanding of already pretrained models.

**Strengths:**

1. **Clear writing:** the paper is well-structured, clearly written, and easy to follow
2. **Method simplicity and effectiveness:** the proposed method is simple, efficient, and shows promising performance improvements across tasks.
3. **Comprehensive experiments:** the experiments cover multiple training settings and analyses, and consistenly demonstrate the method's advantages.

**Weaknesses:**

1. **Limited model scale**: the experiments are conducted only on 50M-parameter models and GPT-2. While these show reasonable performance on simple language tasks (BLIMP & ARC), it remains unlearn whether the improvements generalize to larger models. It would be valuable to see results on larger (1B/7B) models. I understand such experiments are complex and do NOT expect them for the rebuttal.

2. **Limited evaluation scope**: the current evaluations focus mainly on artificial tasks such as word games and digit addition. It would strengthen the paper to provide more results and discussions on  improvements on more realistic tasks.

**Questions:**

1. The experiments only reports results for STOCHASTOK with p<=0.1, is this an intentional design choice? How does performance change with higher p?

2. Does STOCHASTOK improves model robustness to spelling errors or out-of-vocabulary words?

---

> ### Author Response · Authors · 2025-11-21
>
> We thank the reviewer for advocating the acceptance of our paper and for highlighting the paper’s clarity and the consistent effectiveness of the method. We greatly appreciate your encouraging feedback.
>
> **Weaknesses - Model scale and more realistic tasks:** We agree that results on 1B+ models would be valuable. We are a small academic lab and are therefore grateful that you appreciate the cost of pretraining experiments makes them extremely difficult to run in an academic setting. We are currently working on sourcing compute to allow us to run these and hope to have these for the camera ready version of the paper. While more realistic tasks are difficult to evaluate given the small model size of our current experiments, we plan to evaluate the larger models on a wider set of tasks once trained. In the meantime, we hope that our paper inspires further exploration by larger industrial labs. Thank you for this feedback.
>
> **Q1. Performance with higher p:** In Figure 5 we plot final performance over hyperparameter $p$ up to $p=0.2$ (log scale). We found that performance drops off as p is increased beyond 0.1. Intuitively, $p$ needs to be in the "sweet spot" - high enough that the model sees enough splits to enable subtoken understanding, and low enough that the model sees the full tokens often enough (as we evaluate with the standard tokenizer ie. full tokens, no splitting). Curucially, though we find that performance is not that sensitive to $p$, hence given the results in Figure 5, we chose $p=0.1$ as a nice round value to test our method without significant hyperparameter tuning.
>
> **Q2. Robustness to spelling/OOV:** Thank you for this suggestion. In Figure 4 we show that our method improves performance on spelling-related tasks (the CUTE benchmark, see Table 2), and in Figures 11 and 15 we show that it improves robustness to equivalent tokenizations. We agree that a natural addition would be to test robustness to spelling mistakes. Given the performance in other experiments, we anticipate that stochastok will help, and we will run these experiments on the 1B model if we get compute to train this larger 1B+ model.
>
> ----------------
>
> We thank the reviewer again for their invaluable feedback and suggestions. We hope we have addressed all of your questions. Please let us know if you have any more questions or comments as we would be more than happy to discuss further.

---

> > ### Comment · Reviewer_dC4j · 2025-11-28
> >
> > Thanks for the responses, they address my concerns! In general, I think this is a good paper and shoud be in the literature. I will maintain my score.

---

> > > ### Author Response · Authors · 2025-11-29
> > >
> > > Thank you for your reply and for your continued support of our paper! We really appreciate your encouraging feedback.

---

### Comment · Area_Chair_xNgq · 2025-11-28

Dear Reviewers,

The authors have responded to your reviews. Please engage in the discussion and evaluate the authors’ rebuttal to check whether your comments have been adequately addressed, and determine whether you would like to adjust your evaluations.

Best,

Your AC

---

### Meta-Review · Area_Chair_119r · 2026-01-07

**Summary:**

This is an interesting paper that tackles an important but under-studied problem, with an interesting, simple but "elegant" (per some reviewers) solution. Reviewers were impressed by the comprehensivness of the evaluation. Pre- and post-rebuttal, scores were unanimously positive or positive-leaning. There were a number of minor concerns or clarifications (e.g. limited model scale, difference from BPE, etc.), which I think the rebuttal adequately addressed or clarified. Overall, I find this to be an intriguing work that the community would benefit to know/learn more about.

**Reviewer Concerns:**

(Pls see above)

**Reviewer Scores:**

-- ZDUH and dC4j explicitly maintained their scores of 8.
-- eKHE and HeMs did not respond, but would likely maintain their positive-leaning scores of 6.

---

### Decision · Program_Chairs · 2026-01-26

Accept (Poster)